# Identification and Preference of Game Styles in LaLiga Associated with Match Outcomes

**DOI:** 10.3390/ijerph16245090

**Published:** 2019-12-13

**Authors:** Julen Castellano, Miguel Pic

**Affiliations:** 1Department of Physical Education and Sport, University of the Basque Country (UPV/EHU), Vitoria, 48940 Leioa, Spain; julen.castellano@ehu.eus; 2Motor Action Research Group (GIAM), University of La Laguna, 38200 San Cristóbal de La Laguna, Spain

**Keywords:** match analysis, team sport, key performance indicator, interaction, match outcome

## Abstract

The objective was to model the teams’ styles of play (SoPs) in elite football and relate them to the match result. For this, the twenty Spanish first division teams in the 2016–2017 season were analysed, using nine interaction performance indicators (IRi). A principal component (PC) analysis was applied. From two PCs four SoPs were established: deep or high-pressure defending, and elaborate or direct attack. The SoPs were distributed according to average performance obtained throughout the championship. The connection between the preferred SoP and the final result was estimated. Teams with elaborate offensive styles and teams defensively minded got better results. In addition, most of the teams showed variability in their SoP. The applications of the study are (1) the IRi have served to identify SoP and can be used as a reference to optimize team performance; (2) teams should have a varied SoP repertoire, as well as being prepared to deal with different SoPs; (3) particular player profiles should be connected with the desired SoP when creating the squad and (4) clubs should develop a varied range of SoPs at their academies.

## 1. Introduction

Performance indicators are a combination of variables to help understand performance in competitions [1]. The analysis of competitive performance is a key process in having objective information that provides feedback to the team and unveils rival weaknesses and strengths. Most of the studies [2] have focused on the analysis of offensive variables and performance indicators such as passes, ball possession or shots on goal. Only in some cases [3] defensive behaviours were studied (i.e., positioning of the defensive line at the moment of ball recovery). A third alternative has been the proposal of a repertoire of variables and indicators of both phases of the game, offensive and defensive [4,5], which has helped to describe the variables that can tell the difference between winners and losers or between two professional leagues from the same country [6].

Recently, there has been an interest in investigating the description of the SoP (style of play) used by professional football teams [7,8,9,10]. Valuing the performance of the teams from procedural variables, indicators of third order [11], allows for a better interpretation of their performances, distinguishing it from the final result due to the fact that the latter could be more “contaminated” by chance or arbitral decisions with which to misinterpret bad performances despite having won or having lost unfairly after a good team performance. In this sense, it might be interesting to know the team’s game style [12] to design the tasks with greater precision in order to optimize their performance.

In the study by Fernández-Navarro and colleagues (2016) [8], despite the methodological limitations [10], after implementing the principal component technique, they estimated a total of twelve SoPs in the Spanish first division (LaLiga) and the English Premier League. Using the same technique, in a study of the Chinese Superleague [10], they also distributed the teams into four SoPs (Possession, Set Piece, Counter-attack and Transitional Play), from twenty variables that could be grouped into five factors. Through situational variables [7] (i.e., match status, quality of opposition and venue), the direct style was the most used in the Premier League. The quality of opposition showed influence in all styles of play except counterattack. In the same study, while match status had a significant effect on the eight styles, venue showed a significant effect for all styles except counterattack and maintenance. Therefore, contextual variables had an influence on the different styles of play identified by the authors.

However, despite the fact that these proposals may be of interest, they show a parcelled description of team performance in competition. In these works [7,8,10], only the performance in absolute terms of each team was taken into account (i.e., number of counter-attacks, number of passes, number of shots, etc.). In this sense, it might be wise to consider the performance of both teams to obtain relative data of each team in each game which is much more individualized. The performance of teams is based on the rivals’ performance and quality [13] (i.e., number of shots in target done minus the shots in target received).

For this reason, it would be important to include the performance analysis of the opponent’s interaction [14], being the simultaneous inter-motor skills one of the key features of the logic of football, which supports the need to be included the interactive effects [15]. The variance that explains the performance of teams is greater when taken in relative terms in regards to the opponent [16]. It should therefore be noted the methodological importance of considering the variables from an interactive point of view as a relevant procedure to address the specificity of each confrontation.

Therefore, based on these considerations, the present study proposed the following objectives: (1) to identify team SoP of the Spanish first division *LaLiga* through interaction performance indicators (IRi) and (2) to associate the SoPs to the final outcome of the match.

## 2. Methods

### 2.1. Sample

For the elaboration of this study, 373 games (of 380 of the total of the championship) played in the 2016–2017 season of the Spanish first division (*LaLiga*) were analysed. For reasons unrelated to the investigation (i.e., technical errors), seven matches could not be included in the study. The records of the twenty teams involved in the competition gave rise to a total of 731 team performances. The data has been treated in accordance with the Declaration of Helsinki, having been granted the consent of the club to access the data and having received permission from the Ethics Committee on Human Beings (CEISH) of the University of the Basque Country (UPV/EHU).

### 2.2. Variables and Interaction Performance Indicators

The nine IRi were configured from 14 variables, which included offensive, defensive and conditional behaviours of the teams. These variables collect information about the four moments of play [17], that is, ball possession and nonpossession of the ball, as well as its transitions. Most of them have been used in previous studies [4,6,18], proving valid to discriminate performances, although not in the actual version of interaction.

The offensive performance variables were the total percentage of possession, as well as possession at the own field and the rival field, similar to that proposed in previous studies. In addition, the total number of attacks and counterattacks carried out by teams were also included. Finally, total dribbling and successful dribbling were also included, as well as the total number of passes made and those passes made forward.

Among the variables that were collected, the game performance in the defence phase was included, both the total ball recoveries and those made in the opponent’s half [8]. In addition, as has been previously used [18], the average height of the defence was also considered. Finally, as a conditional variable, the total distance covered by the team was considered.

The performance variables of the team were converted into percentage values. From these variables, the IRi was configured, for which the subtraction between the IR value of one team and the IR of the other was made for the match in which they faced each other (Table 1). In this sense, data more adjusted to the specificity of each match was analysed, as opposed to the procedure of adding all the performance values of each team, independently of the performance of the rival team and the specific match.

### 2.3. Procedure

The data was obtained using InStat^®^ (www.instatfootball.com) and Tracab^®^, the latter managed by the application Mediacoach^®^ (http://mediacoach.es/). The reports were exported into Excel (Microsoft Corporation, Redmond, WA, USA), and a matrix made and later analysed. Both types of data offered by these companies in professional football have already been used in previous works [6,9,10], the event system has given reliable figures [9]. Furthermore, using the Bland−Altman method, some previous agreement between measures was done comparing Opta^®^ versus InStat^®^ data. The results showed a bias or systematic error of 5.6 passes (Confidence Intervals 95%: 4.85/6.42; Standard Deviation 479.90 ± 112.4/474.26 ± 113.7), upper (27.1 passes, CI 95%: 25.7/28.4) and lower (−15.8 passes, CI 95%: −17.1/−14.4) limits of agreement; and for recoveries, the systematic error was −0.8 recoveries (CI 95%: −1.31/−0.25; SD 53.06 ± 8.4/53.84 ± 9.1) with upper (13.8 recoveries, CI 95%: 12.9/14.7) and lower (−15.3 recoveries, CI 95%: −16.2/−14.4) limits of agreement. Through the application of intraclass correlation coefficient (95%); “good” results were achieved for recoveries (0.787) and “excellent” for passes (0.997).

For all the analyses, the statistical package IBM SPSS Statistics v24.0 for Windows (SPSS, Chicago, IL, USA) was used. After confirming that the data complied with the criterion of normality applying Shapiro−Wilks [19], acceptable levels of statistical significance were established (*p* < 0.05).

### 2.4. Data Analysis

First, the mean values of the nine IRi were calculated for each of the teams and classified in quartiles. A principal component analysis (PCA) with orthogonal rotation (*Varimax*) was implemented from the mean values. This technique allows to reduce the data, grouping the indicators into a smaller number of components [8,9]. In this way, the IRi with higher values in each component were chosen to distinguish the SoPs given in professional football. The value of Kaiser−Meyer−Olkin (KMO) was calculated to verify the suitability of the sample [20], this value being >0.5. The Bartlett’s sphericity test was significant (*p* < 0.001). The principal axis method was used to extract the components [8,9]. Components with a self-value less than 1 were not preserved for extraction [21]. This is due to the notion that any component that shows a self-value greater than 1 represents a proportion of variance greater than that provided by any variable. The PCA was applied with a *Varimax* rotation to identify that the components or factors were not highly correlated. This ensures that each main component provides different information.

To interpret the components, the absolute values of the coefficients were taken. The correlation coefficients were considered [22] as trivial (r < 0.1), small (0.1 ≤ r < 0.3), moderate (0.3 ≤ r < 0.5), long (0.5 ≤ r < 0.7), very long (0.7 ≤ r < 0.9), almost perfect (r ≥ 0.9) and perfect (r = 1). In the present study, only indicators with a value higher than 0.7 (positive or negative) were considered to define this component [23].

The matches were then classified according to the sign and value assigned to each of the previously generated components; thus revealing the defensive and offensive SoP. Multivariate discriminant analyses were applied [4] starting from *p*-value (*p* < 0.05), using the stepwise method to address the identity of the styles using Wilks’ Lambda [24].

Finally, Chi-square statistic was applied, and interpreted from the adjusted residuals (ar) to know the association between the SoPs and the result (win, lose and draw), for the set of SoPs (χ^2^) and for each one of the teams, in particular considering the preferred SoP (χ^2^PS, style of play more used by teams), being 1 when the team used its medium or preferred SoP and 0 when the team did not play its preferred SoP.

## 3. Results

The mean values and standard deviations of the estimated IRi for each team during the whole championship are shown in Table 2. The values were coloured according to the quartile where they were located: dark means that this value is in Q1, dark grey in Q2, light grey in Q3 and white in Q4.

Table 3 shows the self-values of each main component, as well as the explained and accumulated variance. The first two factors explained almost 80% of the total variance.

The result of the orthogonal rotation analysis of the two main components (Table 4) showed the most important IRi in each of the components. Only IRi with values greater than 0.7 were chosen to define the component. In addition, the IRi will show a positive or negative influence on the component. Within the first component, positive %CON (% of counterattacks) and %PAS (% of forward passes) showed a positive value, while %POS (possession percentage) and %DRI (% of successful dribbles) were negative. Regarding the second component, positive IRi of %POSr (% possession in rival half of the field), %REC (% of recoveries in rival half of the field) and %ALT (% of the space that is left behind the defensive line with respect to the field as a whole) and negative value of %POSp (% possession in own half of the field) were presented. In Table 4, the correlations between IRi are found. The variables %POSr and %ALT had a high positive correlation. At the same time, %POSp had a high negative correlation with %POSr and %ALT, as well as %POS that had a high and negative correlation with the percentage of passes made going forward ratio with respect to that of the rival (%PAS).

Figure 1 shows the distribution of the nine IRi in the two main estimated components, only %KM (% of the distance covered) did not provide information in any components. From this, four SoPs were identified. Two SoPs for representing the offensive phase: direct attack or DA versus elaborate attack or EA, and another two SoP for the defensive phase: deep defending or DD versus high-pressure defence or HD. Quadrant I, identified as direct attack and deep defending (DA/DD), is constructed with positive values of component 1 in the %PAS and %CON variables, and negative values of the %POSp variable of component 2 (top-left of Figure 1). Quadrant II (top-right of Figure 1), direct attack and high-pressure defending (DA/HD), would have positive values in the %POSr, %REC and %ALT variables of component 2 and in %PAS and %CON variables of component 1. In quadrant III (bottom-right of Figure 1), elaborate attack and high-pressure defending (EA/HD), variables with positive values in component 2 (%REC, %POSr and %ALT) and negative in component 1 for %POS. Finally, quadrant IV (bottom-left of Figure 1), elaborate attack and deep defending (EA/DD), with negative values in component 2 (%POSp) and in component 1 (%POS). Average values of the IRi obtained in all the matches played during the championship distributed the teams into the four quadrants, as shown in Figure 1. While most teams showed an unequivocal location reference style, teams like La Coruña, Valencia, R. Betis and Granada were placed in the centre of the figure (i.e., close to 0 in the two components); this could be interpreted as not having an SoP.

Through discriminant analysis, Wilks’ Lambda statistics was applied (*p* < 0.001), which measures the deviations produced by groups with respect to total deviations, fluctuating between 1, when there is no discriminant capacity of the variables, and 0, when they are entirely discriminant. The centres of the groups (centroids) were equal, justified by values lower than 0.3, confirming the singularity of each SoP.

Finally, the association between the SoP and the result (win, lose or draw) is shown in Table 5. There were statistically significant differences taking the SoPs as a whole into account (*p* < 0.014; *df* = 6). The teams that played their games in quadrant II (DA/HD) lost (n = 90, *ar* = 3.0), while those in quadrant IV (EA/DD) won more than expected (n = 90, *ar* = 3.0). Only Espanyol showed a connection between the SoP and the result of the match (*p* < 0.005). Of all their matches (n = 38) when they did not use their preferred SoP (n = 25) they only won six games (*ar* = −2.7) and lost twelve (*ar* = 3.0). However, when they played their preferred style (quadrant I) in fifteen matches they won nine times (*ar* = 2.7) and did not lose any (*ar* = −3.0). Regarding the preferred style, it should be highlighted that in most of the matches, the teams played in the nonpreferred quadrants (n = 428, 57.2%), compared with those played in the preferred style (n = 318, 42.5%), although there was an association between the use of a preferred style and the result of the match (*p* < 0.008; *df* = 6). There was a particular negative connection when using the preferred SoP, a draw (n = 67, *ar* = −1.4) and losing (n = 109, *ar* = −1.9) and positive with winning (n = 142, *ar* = 3.1).

Table 5 distribution of wins, losses and draws depending on each team’s style of play.

## 4. Discussion

The objective of the present study was to model the Spanish first division teams’ SoPs based on interaction performance indicators (IRi), to later evaluate their connection with the match outcomes. The analysis of principal components explained almost 80% of the total variance, allowing to distinguish four SoPs (DA/DD, DA/HD, EA/HD and EA/DD), and which was the predominant or referent (taking the average value) for each team of *LaLiga*. However, it should be noted that the strategic proposal of the teams varied during matches. The second conclusion of the study was that the elaborate attack style (quadrant IV) had a greater association with winning (n = 90, *ar* = 3.0), just the opposite of what happened in matches where teams used quadrant II, whose IRi values are opposed to those of quadrant IV. In addition, the preferred style of each team showed a greater correlation with success (*p* = 0.008).

Unlike previous works on the Spanish League, in two different seasons, 2006–2007 and 2010–2011 [8], the Chinese Super League in the 2016–2017 season [10] and the Greek Superliga in the 2013–2014 season [9], where from 19, 20 and 62 variables; 6, 5 and 8 factors or dimensions were obtained, resulting in 12, 4 and 8 SoPs, respectively. In the current work, with nine IRi only two components or factors have been obtained, from which four SoPs have been proposed. It should be underlined that with only two factors almost 80% of the variance was explained, however, in the same previous studies the first two components reached an explanatory power [8,9,10] of 54%, 40% and 52%, respectively. The IRi have made a more clarifying modelling possible, probably because of the methodological approach, through the interactive relationships of confrontation in competition [15]. In this exploratory model that has been proposed from the nine IRi, it should be noted that %KM had no weight in any component so it was not representative of any SoPs, highlighting, probably, conditional aspects are not so relevant in team sports such as football [25].

In quadrant I, the teams (i.e., Espanyol, Osasuna and Alavés) deployed an SoP DA/DD (Direct/Deep in attack/defence, respectively), using in particular %PAS and %CON, with greater possession in their half than the rival (%POSp) with less dribbling (%DRI). Considering offensive sequences that are short and direct, that is, quick attacks with a low number of passes [1,2], are more effective than longer possessions, the SoP of the teams placed in this quadrant try to exploit their strengths. Nevertheless, the effectiveness of this type of attack has a high connection with the place and the context where the ball is recovered, that is, starting in more offensive zones and in favourable interactional contexts appears to be more effective [2].

The teams found in quadrant II played with a SoP DA/HD (Direct/Pressure). These SoPs showed higher values than those of their rivals in %PAS and %CON for some teams (i.e., Sporting, Leganés, Málaga and Atlético), while others (i.e., Eibar and Athletic) also added superiority in %REC, %ALT and %POSr (IRi that configure component 1). The positioning of the defensive line is a factor that has already been previously studied [18], proposing a predictive model based on linear regression, through the advance of defensive and offensive lines when the team had less quality, played at home and was losing on the scoreboard [3]. This factor had been studied to compare the teams placed in the upper half of the table with respect to those in the lower half of two professional leagues of the same country [6]. This variable showed a certain degree of sensitivity, the best-ranked teams had their defence high, although it may be because the best-ranked teams also had more ball possession [7] and, therefore, were most of the time in the attack phase. Recently, it has been found [2] that regaining possession in more offensive zones through pressing could improve the effectiveness of the offensive phase. On the other hand, although some of the teams with this SoP (i.e., Sporting and Leganés) showed no interest in using an elaborate style, but they also showed that they covered more distance than their rivals did.

Placed in quadrant III, only three teams (R. Sociedad, R. Madrid and Barcelona) showed an SoP EA/HD (Elaborated/Pressing), where in addition to higher values than their rivals in the IRi of %POS, %ALT, %POSr and %REC, showed lower values with respect to their opponents in %CON and %PAS. Recovering the ball closer to the opponent’s goal increases the chance of scoring [26]. In addition, although the %KM indicator has not been part of any of the components, these teams could have a lower physical requirement associated with high-pressure defending [27]. It would be in a sense logical that the use of EA/HD would require playing in smaller spaces and therefore, the importance of %KM could be relativized. Thus, if something distinguishes this style, it would be the high importance of the synchronization [28] and the great technical repertoire of the players [29], compared with styles in which a greater distance could be more valued (i.e., Espanyol, Atlético, Sporting, Leganés and R. Betis, in the current study).

Finally, in quadrant IV, where teams are grouped with Elaborate Attack and Deep Defending (EA/DD), they had higher values than their rivals in %POS and were lower in variables such as %REC and %ALT. The SoP that is placed in this quadrant could be understood as those teams that even having had control of the ball have not had control of the match. In spite of having had more possession than their rivals, they have done so in their own half, hence the offensive-progression index has not benefitted, this is understood as pass efficiency (i.e., ratio shots/passes), which is considered a key aspect in the good performance of a team [30].

Similar to that found in previous works [8,10], this study points out the importance of variability in the styles of play used by teams during competition. Although the average values in the IRi placed each team in a single quadrant or preferential SoP, the reality of the interaction forced the teams to adapt to the context, that is, level of the opponent, location, current score, and so forth. [9]. The intra-team variability shown during the championship could be explained by the property of degeneracy or redundancy of a complex adaptive system when the teams behave in a collective duel as is football. This redundancy comes to represent the idea that structurally different components of the system (i.e., changes to team line-up throughout the championship) like players that can play a similar role—even not being identical—with respect to the context. This is a synergistic feature of team behaviour, where the rival counts, mainly, needing to adapt to the particularity of that match [31], having repercussions on the variability experienced by players depending on the match, for example, in the physical demand. The results of the present study suggest that there is a strategic coherence in team SoP, where each preferential behaviour is located. We focus our attention on the average values, but the variability in the SoP used by the teams during the championship must not be forgotten, sometimes as a result of the coach dismissal. While there is a particularity in the teams’ style of play, arising from a source of strategic order pre-established and agreed between the coaching staff and players, during the match it may be changed for different reasons that contextualize the teams “needs” in that particular moment. This variability in the SoPs highlights the need to reorient the organization of the training process to develop a unique style of play, one’s own (i.e., consolidation of strategic fidelity to optimize a unique way of competing), towards the need to prepare the team to be able to develop flexible, adaptable and varied SoPs to deal with the particular dynamics intra- and inter-matches (i.e., deal with different rivals or changes in the score, sending-offs, etc.).

A relevant aspect of the study is the study of the association between the SoP deployed during the competition by teams and match outcome. When teams did not play their preferred SoP they also won (or tied) and, on the contrary, lost when they played in the style that they know “best”. The reality is that most teams deployed a nonpreferred style, 56.4% compared with 42.6% who did, either because they decided or because they were forced to do so. Only Espanyol offered a significant connection (*p* = 0.005) between their preferred SoP (quadrant I) and victories, of 15 games played in their style they won nine (*ar* = 2.7) and lost none (*ar* = −3).

With the exception of Atlético (with a SoP DA/HD), the most successful teams in *LaLiga* (fourth quartile of the points distributed in the championship, with >64 points, the top six that gave them the possibility to play the following year in Europe), were located in quadrants III (EA/HD) and IV (EA/DD). The IRi that best represented the SoPs were ball possession (%POS, and %POSr with positive values and %POSp with negative values) and a deep defensive line (%ALT). Probably with the intention of avoiding a comeback or once ahead on the scoreboard. The ball possession indicator is important depending on the league, although it is associated with success in the Bundesliga [32], Spanish League [6] and Premier League [33], or in international tournaments such as the World Cup [34]. Furthermore, it seems that football is evolving towards the possession game, due to an increase in the number and efficiency in passing [35], mainly in players on their defensive line [36].

In the lower part of the table, teams that obtained the first quartile of the distributed points (<39 points), from the last placed to the 14th position, showed a direct attack profile and deep defending style, with the exception of Las Palmas (quadrant IV). Counter-attacks and/or direct play are more effective in reaching the opponent’s area than an elaborate attack, especially playing against unbalanced defences [26]. Quadrants III and IV can be linked to taking the initiative in the offensive game, however, not being able to overcome the last defensive line [14], inefficiency in shooting [6] and other variables to be discovered could explain low yields even in these quadrants (i.e., Las Palmas and Celta placed 13th and 14th, respectively).

One of the limitations of this study has to do with the fact that SoPs respond to the variables and indicators that have been chosen for the present study. Probably, the choice of other variables, or the incorporation of new ones (i.e., duels, long passes, rejections, crosses to the area, etc.), could incorporate nuances to the SoP and, therefore, could refine the profile description of the team performance. The third limitation has to do with how to consider success (i.e., match outcomes: winning, drawing or losing), and the performance, average match IRi value just for a single season. The problems implicit in these elections are two: first, that good performance and winning do not always go together, although the fact of having analysed the performance of almost all the matches could mitigate it, and; second, because based performance on the average IRi value once completed could conceal alternating dynamics in the teams’ SoPs as a consequence of the situation variables (i.e., as a consequence of leading the ranking). Literature supports the need to consider the temporary score [13] as a source of variability in both behavioural and in the conditional performance. In this sense, incorporated the intra-match periods into the analysis based on the current score could have reflected different game strategies from those given by the accumulated values at the end of the match. Finally, together with the previous one, it would have been interesting to consider a greater number of contextual variables [9] (i.e., location, quality of the rival, etc.) that are known to condition the strategic proposals of the teams [7,31]. For this reason, in the future it would be interesting to apply this same SoP analysis taking into account the possible contextual variables that condition the development of the game, while knowing the effectiveness that these approaches have in relation to the score in that moment.

## 5. Conclusions

The present study reaches two main conclusions. In the first place, it has been possible to identify four different SoPs in *LaLiga* from nine IRi and associate a preferential SoP for each team, although the vast majority of the teams had to propose different strategic proposals during the championship. Second, not all SoPs were equally associated with success in that match (i.e., winning, drawing and losing), although preferential styles brought teams closer to success. The applications of the study are (1) the IRi have served to identify SoPs and can be used as a reference to optimize team performance; (2) teams should have a varied SoP repertoire, as well as being prepared to deal with different SoPs; (3) particular player profiles should be connected with the desired SoP when creating the squad; and finally; (4) clubs should develop a varied range of SoPs at their academies.

## Figures and Tables

**Figure 1 ijerph-16-05090-f001:**
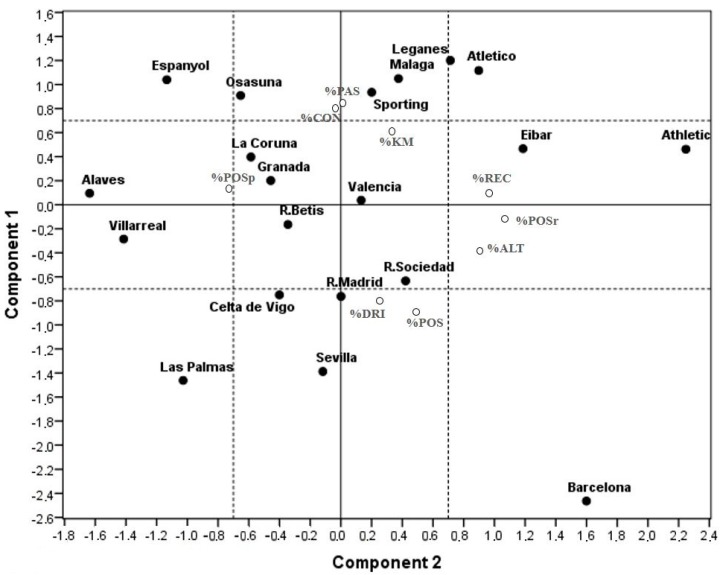
IRi distribution as the average position of each team through the two principal components. Note: %POS is possession percentage, %POSp is % possession in own half of the field, %POSr is % possession in rival half of the field, %CON is % of counterattacks, %REC is % of recoveries in rival half of the field, %PAS is % of forward passes, %DRI is % of successful dribbles, %ALT is % of the space that is left behind the defensive line with respect to the field as a whole, and %KM is % of the distance covered.

**Table 1 ijerph-16-05090-t001:** Definition of variables, formulas and codes of performance indicators in interaction (IRi).

Definition	Formula	IRi Code
Difference in the percentage of ball possession (POS) between teams	((POS total [A]/(POS total [A] + POS total [B])) *100) − ((POS total [B]/(POS total [A] + POS total [B])) *100)	%POS
Difference in the percentage of ball possession in own half of the field with respect to the total between teams	((POS own half of the field [A]/POS total[A])*100) − ((POS own half of the field [B]/POS total[B])*100)	%POSp
Difference in the percentage of ball possession in rival half of the field with respect to the total between teams	((POS rival half of the field [A]/POS total [A])*100) − ((POS rival (half of the) field [B]/POS total [B])*100)	%POSr
Percentage of counterattacks with respect to the total attacks (without dead ball actions (BP)), understanding as counterattacks the attacks with a maximum duration of at least 20 s and in which the ball advances at more than 3 m/s	((Counterattacks [A]/nº of total attacks [A]) *100) − (Counterattacks [B]/nº of total attacks [B]) *100)	%CON
Percentage of recoveries in rival (half of the) field (losses of the rival team in own field) with respect to the total recoveries (total losses of the rival team)	((Recoveries in rival half of the field [A]/Total Recoveries [A])*100) − ((Recoveries in rival half of the field [B]/Total recoveries [B])*100)	%REC
Percentage of forward passes with respect to total passes by the team	((Forward passes [A]/Total passes[A]) *100) − ((Forward passes [B]/Total passes [B]) *100)	%PAS
Percentage of successful dribbling and total dribbling by the team (A or B)	((Total dribbles [A]/(Total dribbles[A]+ Total dribbles [B]))*100) − ((Total dribbles [B]/(Total dribbles [A]+ Total dribbles [B]))*100)	%DRI
Percentage of space between the last defence (goalkeeper is not included) and the goal line	((spatial depth of defense [A]/100) *100) − ((spatial depth of defense [B]/100) *100)	%ALT
Percentage of distance covered (KM) by the team with respect to the distance covered by both teams	((KM [A]/(KM [A]+ KM [B])*100) − ((KM [B]/(KM [A]+ KM [B])*100)	%KM

**Note:** [A] is team A and [B] is team B.

**Table 2 ijerph-16-05090-t002:** Average values (standard deviation) of interaction performance indicators for each team in *LaLiga*.

Equipo	%POS	%POSp	%POSr	%CON	%DRI	%REC	%PASES	%KM	%ALT
R. Madrid	11.2(14.9)	−1.0(14.3)	1.0(14.3)	0.2(8.1)	9.3(15.3)	1.0(13.4)	−1.6(3.8)	**−1.8(1.8)**	3.3(6.2)
Barcelona	29.8(15.6)	**−8.8(17.6)**	8.8(17.6)	**−7.5(8.1)**	26.1(14.5)	4.3(16.6)	**−6.5(5.3)**	**−1.1(1.9)**	7.0(8.3)
Atlético	−4.2(18.8)	−2.5(12.7)	2.5(12.7)	1.7(9.5)	−0.7(19.0)	1.5(11.5)	1.6(5.0)	1.6(1.6)	−0.1(6.3)
Sevilla	14.8(15.2)	−1.7(12.8)	1.7(12.8)	**−4.2(8.7)**	3.3(20.7)	**−3.2(13.0)**	**−5.4(4.5)**	0.2(1.8)	0.1(6.0)
Villarreal	−2.2(18.9)	7.8(12.8)	**−7.8(12.8)**	0.1(8.4)	−3.1(18.3)	**−4.7(12.4)**	−1.8(5.0)	−0.6(1.5)	**−2.8(7.2)**
R. Sociedad	12.8(13.7)	−2.3(12.8)	2.3(12.8)	**−3.2(7.3)**	4.0(16.2)	3.7(13.6)	**−2.3(4.2)**	0.1(1.6)	2.2(5.4)
Athletic	3.9(15.7)	**−8.6(11.0)**	8.6(11.0)	0.9(8.5)	−4.1(23.5)	10.1(10.4)	−0.7(4.8)	0.4(1.7)	5.2(4.9)
Espanyol	**−13.7(15.5)**	6.2(13.3)	**−6.2(13.3)**	4.6(8.1)	**−7.6(21.0)**	**−7.6(10.5)**	3.6(4.6)	2.4(1.5)	**−4.3(4.5)**
Alavés	**−8.3(18.5)**	9.9(13.3)	**−9.9(13.3)**	2.9(8.1)	−4.0(17.0)	**−4.9(10.8)**	1.2(5.6)	0.2(1.5)	**−5.4(5.7)**
Eibar	0.1(16.6)	**−5.0(11.1)**	5.0(11.1)	1.9(7.2)	**−9.4(18.0)**	9.8(11.3)	0.1(5.3)	0.3(2.0)	2.7(4.4)
Málaga	−6.1(17.2)	**−3.3(15.4)**	3.3(15.4)	3.5(7.7)	−5.4(21.8)	0.6(11.5)	3.4(5.7)	0.5(1.7)	0.2(6.8)
Valencia	−4.0(17.4)	−1.9(13.8)	1.9(13.8)	−1.2(9.2)	3.0(18.3)	0.4(13.1)	0.4(4.7)	0.1(1.9)	0.4(6.3)
Celta	4.8(18.4)	1.3(10.2)	−1.3(10.2)	**−1.8(7.0)**	−4.4(17.8)	0.6(8.9)	**−2.7(6.1)**	**−2.1(1.4)**	**−2.6(4.2)**
Las Palmas	13.1(19.4)	2.6(13.6)	**−2.6(13.6)**	**−6.9(10.9)**	16.4(17.5)	**−9.8(15.9)**	**−3.5(5.2)**	**−0.9(1.7)**	−0.4(7.2)
R.Betis	0.1(16.0)	2.3(11.8)	−2.3(11.8)	1.0(7.8)	−1.6(15.0)	0.2(11.6)	−0.9(4.5)	0.6(1.8)	−0.4(5.5)
La Coruña	−6.1(18.4)	2.3(14.0)	−2.3(14.0)	2.3(10.3)	−2.3(20.4)	**−3.1(9.3)**	1.4(5.0)	**−1.0(1.4)**	**−1.9(6.6)**
Leganés	**−10.9(15.6)**	**−4.3(11.9)**	4.3(11.9)	1.7(7.8)	−9.0(16.9)	0.2(9.9)	2.7(5.3)	0.9(1.6)	0.2(5.0)
Sporting	**−9.9(16.6)**	0.8(10.3)	−0.8(10.3)	−0.5(6.4)	**−10.4(18.9)**	4.7(8.1)	4.3(6.1)	1.1(1.6)	−1.0(5.8)
Osasuna	**−16.8(14.5)**	4.1(15.2)	**−4.1(15.2)**	4.2(8.9)	**−7.9(15.8)**	−1.7(9.9)	4.4(5.3)	−0.3(1.8)	−1.0(6.2)
Granada	**−8.2(18.4)**	2.0(13.4)	−2.0(13.4)	0.3(8.0)	8.1(19.6)	−2.0(12.1)	2.3(5.5)	**−0.9(2.0)**	−1.2(4.9)

**Note:** %POS is possession percentage, %POSp is % possession in own half of the field, %POSr is % possession in rival half of the field, %CON is % of counterattacks, %REC is % of recoveries in rival half of the field, %PAS is % of forward passes, %DRI is % of successful dribbles, %ALT is % of the space that is left behind the defensive line with respect to the field as a whole and %KM is % of the distance covered. The values were coloured according to the quartile where they were located (dark means that this value is in Q1, dark grey in Q2, light grey in Q3 and white in Q4).

**Table 3 ijerph-16-05090-t003:** Eigenvalues for components and total variance explained.

Component	Initial Eigenvalues			Extraction Sums of Squared Loadings			Rotation Sums of Squared Loadings		
	Total	% Variance	% Cumulative	Total	% Variance	% Cumulative	Total	% Variance	% Cumulative
1	4.7	51.82	51.82	4.67	51.82	51.82	3.82	42.49	42.49
2	2.5	27.93	79.74	2.51	27.93	79.74	3.35	37.26	79.74
3	0.67	7.44	87.18						
4	0.46	5.10	92.28						
5	0.30	3.34	95.66						
6	0.20	2.25	97.90						
7	0.15	1.66	99.56						
8	0.04	0.43	99.99						
9	0.001	0.01	100.0						

**Table 4 ijerph-16-05090-t004:** Rotated principal components (PC) matrix Varimax, loadings of each IRi (indicator of each interactive performance) and Pearson correlation matrix between IRi.

IRi	PC		Pearson							
	1	2	%POSp	%POSr	%CON	%DRI	%REC	%PAS	%KM	%ALT
%POS	−0.910	0.342	−0.42 ^#^	0.42 ^#^	−0.62 ^#^	0.56 ^#^	0.25 ^#^	−0.83 ^#^	−0.29 ^#^	0.67 ^#^
%POSp	0.126	−0.959		−1.00 ^#^	0.27 ^#^	−0.26 ^#^	−0.59 ^#^	0.28 ^#^	0.03	−0.81 ^#^
%POSr	−0.137	0.953			−0.27 ^#^	0.26 ^#^	0.59 ^#^	−0.28 ^#^	−0.03	0.81 ^#^
%CON	0.877	−0.181				−0.37 ^#^	−0.14 ^#^	0.51 ^#^	0.26 ^#^	−0.42 ^#^
%DRI	−0.853	0.102					0.06	−0.44 ^#^	−0.28 ^#^	0.37 ^#^
%REC	0.111	0.863						−0.09 *	−0.04	0.65 ^#^
%PAS	0.931	−0.170							0.27 ^#^	−0.44 ^#^
%KM	0.657	0.174								−0.12 ^#^
%ALT	−0.390	0.750								

**Note:** %POS is possession percentage, %POSp is % possession in own half of the field, %POSr is % possession in rival half of the field, %CON is % of counterattacks, %REC is % of recoveries in rival half of the field, %PAS is % of forward passes, %DRI is % of successful dribbles, %ALT is % of the space that is left behind the defensive line with respect to the field as a whole, and %KM is % of the distance covered. * *p* < 0.05, ^#^
*p* < 0.01.

**Table 5 ijerph-16-05090-t005:** Distribution of wins, losses and draws depending on each team’s style of play.

			Average Quadrat	I			II			III			IV				
Team	Ranking	Pts		D	W	L	D	W	L	D	W	L	D	W	L	*p*(*χ*^2^)	*p*(*χ*^2^PS)
R. Madrid	1	93	III	1	3		3	1	1	1	**10**	1	1	13	1	0.095	0.632
Barcelona	2	90	III							4	**17**	4	2	11		0.301	0.301
Atlético	3	78	II	2	8	1	3	**8**	5	4	3			4		0.089	0.083
Sevilla	4	72	IV		1	2	1	2	2	4	8	1	4	**10**	3	0.356	0.887
Villarreal	5	67	IV	3	4	3	1	2	1	3		2	3	**13**	3	0.222	0.075
R. Sociedad	6	64	III		2		2	2	3	3	5	5	2	10	4	0.539	0.590
Athletic	7	63	II		4	2	1	**10**	6	5	4	4				0.112	0.248
Espanyol	8	56	I	4	**9**	8	4	5	3			1	3	1		0.247	0.005
Alavés	9	55	I	5	**11**	4	2	1	1	1		1	5	2	4	0.421	0.065
Eibar	10	54	II	2	4	2	2		5	4	2	6	1	2	1	0.631	0.483
Málaga	11	46	II	4	4	5	4		5	2	3	5				0.247	0.894
Valencia	12	46	II	2	4	2	1		**7**	4	3	2		2	6	0.097	0.281
Celta	13	45	IV	1	1	4		1	**5**	1	2	5	4	**8**	7	0.768	0.248
Las Palmas	14	39	IV		1	2			**3**	5	2	5	4	**6**	11	0.635	0.629
R.Betis	15	39	I	2	**5**	5	2	1	5	2	2	6	3	2	3	0.726	0.336
La Coruña	16	36	I	6	**5**	6	2	1	2	1	2	5	2		4	0.629	0.376
Leganés	17	35	II	6	2	2	3	**3**	11	1	1	4	1	2	2	0.203	0.248
Sporting *	18	31	II	2	1	5	6	**4**	9			6	2	2		0.097	0.699
Osasuna *	19	22	I	5	**3**	7	3	1	12	1		2	1		3	0.666	0.162
Granada *	20	20	I	2		4	1	2	8	1		7	3	2	7	0.586	0.455
			Total	46	72	64	41	41	94	47	64	72	41	90	59	0.014	0.008

**Note:** L is losing, D is drawing and W is winning. * Relegated teams. Ranking is the position in the classification at the end of the season. Average quadrant refers to the location of the mean value of the two components; *p*-values (χ^2^, is the chi-square between the result achieved by the teams in each quadrant, while *p*-values (χ^2^PS, chi-square for the preferred style of play) reflects the association between the result and the preferred style. The victories that were achieved in their preferred style of play (quadrant) are bold.

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
