# Peer review of "Identification and Preference of Game Styles in LaLiga Associated with Match Outcomes"

_ijerph, 2019, doi:10.3390/ijerph16245090_

Round 1

Reviewer 1 Report

Dear authors, first of all I would like to congratulate you for the maniscrip, which without any doubt is original.

Reviewer Comments

The current study aimed to identify team SoP of the Spanish first division LaLiga through interaction performance indicators (IRi), and, 2) to associate the SoPs to the final outcome of the match. The article has merit. The proposal to consider the rival in the analysis and express the indicators in percentage value is very interesting However, some minor revisions must be done before acceptance. Please consider the following specific comments.

Introduction:

In line 1 of the introduction (line 24) it is indicated that: Performance indicators are a combination of variables to help understand performance in  competitions, and subsequently and directly analyzes them in offensive, defensive or a combination of them to analysis performance. Are there no other indicators that are only of tactical behaviors of the team? Is this common to all sports?

In line 33:  The term SoP is not defined.

Methods

In line 86 autors said: “The offensive performance variables were: the total percentage of possession…” Is the total percentage of possession always an offensive variable?

In line 83: The authors consider the total distance as a conditional variable, but this conditional variable is one of the least related to performance. Why use this variable and not other vaiables such as the distance covered at high intensity?

In line 108-109: “Furthermore, using the Bland-Altman method 108 some previous agreement between measures were done comparing Opta® versus InStat® data.” Is the use of Opta® correct?

Results

In table 3: What are the two main components of the table?

The preferred style of each team is conditioned by the level of the opposing team. The first teams of the classification will keep it throughout the year, but will the latter not be more conditioned by the opponent than by their preferences?

Discussión: In line 15 the authors say: “… it should be noted that %KM had no weight in any  component so it was not representative of any SoPs, highlighting, probably, conditional aspects are not so relevant in team sports such as football” All conditional components or just the total distance covered?

Author Response

Response to Reviewer 1 Comments

Dear authors, first of all I would like to congratulate you for the maniscrip, which without any doubt is original.

Reviewer Comments

The current study aimed to identify team SoP of the Spanish first division LaLiga through interaction performance indicators (IRi), and, 2) to associate the SoPs to the final outcome of the match. The article has merit. The proposal to consider the rival in the analysis and express the indicators in percentage value is very interesting However, some minor revisions must be done before acceptance. Please consider the following specific comments.

Introduction:

In line 1 of the introduction (line 24) it is indicated that: Performance indicators are a combination of variables to help understand performance in  competitions, and subsequently and directly analyzes them in offensive, defensive or a combination of them to analysis performance. Are there no other indicators that are only of tactical behaviors of the team? Is this common to all sports?

Response: These questions are very relevant, then; when their reliability, validity and accuracy have been tested, tactical behaviors can be considered performance indicators, because they are connected with performances and discriminating successful and unsuccessful teams. Second, as far as we know, and in our opinion, it is common in all sports.

In line 33:  The term SoP is not defined.

Response: Thank you very much, it is introduced in the text.

Methods

In line 86 autors said: “The offensive performance variables were: the total percentage of possession…” Is the total percentage of possession always an offensive variable?

Response: In the football context, it is considered the attack phase since that the ball belong to the team, however sometimes the team with ball does not want to attack. But this fact, happens rarely so it is standardized that the possession is related to the attack phase.

In line 83: The authors consider the total distance as a conditional variable, but this conditional variable is one of the least related to performance. Why use this variable and not other variables such as the distance covered at high intensity?

Response: Total distance was used as a conditional variable because it is the variable that is related to the purely energetic aspect in our study, but for sure other variables such as  the high intensity distance, numbers of acceleration and deceleration activities would be a very interesting variable to consider in future studies. The limitations of access to this type of data will also be completed in future studies.

In relation to the above, indeed; % KM had a relative importance in the formation of the 4 game styles. Thank you very much for these contributions.

In line 108-109: “Furthermore, using the Bland-Altman method 108 some previous agreement between measures were done comparing Opta® versus InStat® data.” Is the use of Opta® correct?

Response: The quality of the data was calculated by connecting the study of variables measured by two different companies. We cannot know which data is of better quality, but we can affirm that both systems offer very similar data, which could mean that the data is reliable.

Results

In table 3: What are the two main components of the table?

Response: Components 1 and 2. It can be seen in table 3, with almost 80% of the variance explained. The content of each component involve, with different weight (importance), a combination of the variables that appear in the table 4. 

The preferred style of each team is conditioned by the level of the opposing team. The first teams of the classification will keep it throughout the year, but will the latter not be more conditioned by the opponent than by their preferences?

Response: This indication is very important. Due to this, we include the teams quality as a limitation of the study. We add a quote to reinforce its importance this limitations.

Discussión: In line 15 the authors say: “… it should be noted that %KM had no weight in any  component so it was not representative of any SoPs, highlighting, probably, conditional aspects are not so relevant in team sports such as football” All conditional components or just the total distance covered?

Response: Indeed, none of the components was highly conditioned by the %KM variable, the weights of the first and second components being: .657 and .174; respectively. In this study, %KM is the only conditional variable. Thank you very much for this suggestion.

Reviewer 2 Report

L108, “Furthermore, using the Bland-Altman method 108 some previous agreement between measures were done comparing Opta® versus InStat® data. The 109 results showed a bias or systematic error of 5.6 passes (IC95%: 4.85/6.42), upper (27.1 passes, IC95%: 110 25.7/28.4) and lower (-15.8 passes, IC95%: -17.1/-14.4) limits of agreement; and for recoveries the 111 systematic error was -0.8 recoveries (IC95%: -1.31/-0.25) with upper (13.8 recoveries, IC95%: 12.9/14.7) 112 and lower (-15.3 recoveries, IC95%: -16.2/-14.4) limits of agreement.” Please show the standardized typical error and their ICs. Table 1, the formula “((spatial depth of defense [A]/100) *100) - ((spatial depth of defense [B]/100) *100)” equals “spatial depth of defense [A] - spatial depth of defense [B]”, how could it generate a percentage value? Table 5, the χ2 showed here is the significant value (p value) rather than the chi-square value of the chi-square test. Please rephrase these sentences:

L38. By focusing attention on metrics with which to assess the procedures of ball possession, non-possession, transitions between them and set-pieces [12], would allow to have the relevant information on which to work on during the week with the team to optimize their performance.

L47. In a recent work [7], and incorporating certain situational variables, (i.e., match status, quality of opposition and venue), found that the direct style was the most used in Premier League by visiting weaker teams in the 2015-2016 season.

L58. That is, consider the performance of teams based on the rivals' performance and quality [13], i.e. number of shots in target done minus the shots in target received.

L61. For this reason, it would be important to include the performance analysis of the opponent’s interaction [14], being the simultaneous inter-motor skills one of the key features of the logic of football, which supports the need to avoid the interactive effects that occurred in a football match [15].

L211 However, when they played their preferred style (quadrant I) in fifteen matches they won nine times (ar = 2.7) and did not lose any (ar = -3.0). However, when they played their preferred style (quadrant I) in fifteen matches they won nine times (ar = 2.7) and did not lose any (ar = -3.0).

L255 “not using dribbling (%DRI)”. Using fewer dribbling than opponent does not mean no using dribbling at all.

L281 “use SoP EA / HD reduce the space available to play and therefore, the relative importance of %KM”.

Author Response

Response to Reviewer 2 Comments

L108, “Furthermore, using the Bland-Altman method 108 some previous agreement between measures were done comparing Opta® versus InStat® data. The 109 results showed a bias or systematic error of 5.6 passes (IC95%: 4.85/6.42), upper (27.1 passes, IC95%: 110 25.7/28.4) and lower (-15.8 passes, IC95%: -17.1/-14.4) limits of agreement; and for recoveries the 111 systematic error was -0.8 recoveries (IC95%: -1.31/-0.25) with upper (13.8 recoveries, IC95%: 12.9/14.7) 112 and lower (-15.3 recoveries, IC95%: -16.2/-14.4) limits of agreement.” Please show the standardized typical error and their ICs.

Response: We are sorry, but we do not know exactly what values you are referring to. We believe that the values necessary to be able to correctly interpret the results of B-A are in the table below, as has been done in previous works. In any case, we show the example of these two variables, with the average values and standard deviations of each system, as well as the bias, the bias standard error and their confidence intervals.

Variable

Opta

SD

Instat

SD

Bias

ES (bias)

IC 95% ES

passes

479.90

±112.4

474.26

±113.7

5.64

0.40

4.85/6.42

rec

53.06

±8.4

53.84

±9.1

-0.78

0.27

-1.31/0.25

Table 1, the formula “((spatial depth of defense [A]/100) *100) - ((spatial depth of defense [B]/100) *100)” equals “spatial depth of defense [A] - spatial depth of defense [B]”, how could it generate a percentage value?

Response: It could be better to understand through an example: if the defense is 45 m from its goal with respect to the 100 meters that measures the length of the field, it means that it is in 45% of the field (if 100 m is 100%, how much is 45? = (45m / 100m) * 100) = 45%. The formula of both teams, one that has the defense at 45m and another at 50m would be this: (45m / 100m) * 100- (50m / 100m) * 100 = 45% -50% = - 5%

Table 5, the χ2 showed here is the significant value (p value) rather than the chi-square value of the chi-square test.

Response: Thanks for this comment. It has been corrected.

Please rephrase these sentences:

L38. By focusing attention on metrics with which to assess the procedures of ball possession, non-possession, transitions between them and set-pieces [12], would allow to have the relevant information on which to work on during the week with the team to optimize their performance.

Response: It has been modified

L47. In a recent work [7], and incorporating certain situational variables, (i.e., match status, quality of opposition and venue), found that the direct style was the most used in Premier League by visiting weaker teams in the 2015-2016 season.

Response: Thank you very much. It was rectified

L58. That is, consider the performance of teams based on the rivals' performance and quality [13], i.e. number of shots in target done minus the shots in target received.

Response: It was modified

L61. For this reason, it would be important to include the performance analysis of the opponent’s interaction [14], being the simultaneous inter-motor skills one of the key features of the logic of football, which supports the need to avoid the interactive effects that occurred in a football match [15].

Response: It is rectified

L211 However, when they played their preferred style (quadrant I) in fifteen matches they won nine times (ar = 2.7) and did not lose any (ar = -3.0). However, when they played their preferred style (quadrant I) in fifteen matches they won nine times (ar = 2.7) and did not lose any (ar = -3.0).

Response: Thank you very much, the error is eliminated.

L255 “not using dribbling (%DRI)”. Using fewer dribbling than opponent does not mean no using dribbling at all.

Response: Thank you very much for this suggestion. It was rectified.

L281 “use SoP EA / HD reduce the space available to play and therefore, the relative importance of %KM”.

Response: We rectify in the text. Thanks for this comment.
